# Neuronal Growth and Formation of Neuron Networks on Directional Surfaces

**DOI:** 10.3390/biomimetics6020041

**Published:** 2021-06-16

**Authors:** Ilya Yurchenko, Matthew Farwell, Donovan D. Brady, Cristian Staii

**Affiliations:** Department of Physics and Astronomy, Tufts University, Medford, MA 02155, USA; Ilya.Yurchenko@tufts.edu (I.Y.); Matthew.Farwell@tufts.edu (M.F.); Donovan.Brady@tufts.edu (D.D.B.)

**Keywords:** neuron, axonal growth, neuron networks, neural repair, tissue engineering, stochastic processes

## Abstract

The formation of neuron networks is a process of fundamental importance for understanding the development of the nervous system and for creating biomimetic devices for tissue engineering and neural repair. The basic process that controls the network formation is the growth of an axon from the cell body and its extension towards target neurons. Axonal growth is directed by environmental stimuli that include intercellular interactions, biochemical cues, and the mechanical and geometrical properties of the growth substrate. Despite significant recent progress, the steering of the growing axon remains poorly understood. In this paper, we develop a model of axonal motility, which incorporates substrate-geometry sensing. We combine experimental data with theoretical analysis to measure the parameters that describe axonal growth on micropatterned surfaces: diffusion (cell motility) coefficients, speed and angular distributions, and cell-substrate interactions. Experiments performed on neurons treated with inhibitors for microtubules (Taxol) and actin filaments (Y-27632) indicate that cytoskeletal dynamics play a critical role in the steering mechanism. Our results demonstrate that axons follow geometrical patterns through a contact-guidance mechanism, in which geometrical patterns impart high traction forces to the growth cone. These results have important implications for bioengineering novel substrates to guide neuronal growth and promote nerve repair.

## 1. Introduction

Neurons are the basic cells that make up the nervous system. During their growth, neurons extend two types of processes: axons and dendrites, which navigate to other neurons and form complex neuronal networks that transmit electrical signals throughout the body. The extension of the axon is guided by its growth cone, a motile unit located at the distal tip of the axon that navigates through the surrounding environment using electrical, chemical, mechanical, and geometrical cues [1,2,3,4]. The dynamics of the growth cone is controlled by a flexible ensemble of actin and microtubule filaments that form the neuron cytoskeleton [1,2,3,4,5,6,7].

Previous research has identified many of the molecular pathways responsible for intercellular signaling in the formation of neuronal networks [1,2,3,4,5,6,7]. It is now well-established that the biomechanical properties of neurons are an integral part of their functional behavior and play an essential role in normal brain development. For example, it is known that the growing axon is capable of detecting a large variety of biochemical, mechanical, and topographical cues within the growth environment and of directing its growth over relatively long distances (hundreds of microns) with great precision [1,2,3,4,5,6,7,8]. To understand how neurons grow axons and dendrites and wire up the nervous system, we need to understand how they respond to external physical stimuli.

Much of the research into how geometric and mechanical cues affect neuronal growth has been performed in vitro on an ensemble of neurons grown on substrates where the geometry can be controlled. These studies have shown that neurons grown on substrates with periodic geometrical features develop different growth patterns when compared to neurons grown on surfaces lacking a periodic geometry. Such differences include populations of axons that are significantly longer and that tend to align their growth with preferred spatial directions [9,10,11,12,13,14,15,16].

The ability to control and direct neuronal growth in vitro has important consequences for exploiting bioinspired designs for applications in tissue engineering, neural repair, and in vitro–in vivo device interfaces. A major goal in neural tissue engineering is to create controlled biologically inspired environments that promote axonal growth and reproduce the physiological conditions found in vivo [3,5,9,10,11,15,17,18]. However, there are still major challenges with respect to the ability to control and direct neuronal growth. For example, despite recent advances, there are still key unanswered questions about the mechanisms that control neuron biomechanical responses, as well as about the details of cell-substrate interactions, such as the synergy or antagonism between various external cues. Furthermore, most of the previous work on studying neuronal growth in vitro has focused on qualitative or semi-quantitative models to describe the influence of geometrical or mechanical cues on the formation of the neuronal network. A detailed characterization of the basic mechanisms that underlie the growth cone response to physical cues is still missing.

In our previous work, we have shown that axonal growth on surfaces with controlled geometries arises as the result of an interplay between deterministic and stochastic components of growth cone motility [10,15,16,19,20]. Deterministic influences include, for example, the presence of preferred directions of growth along specific geometric patterns on substrates, while stochastic components come from the effects of polymerization of cytoskeletal elements (actin filaments and microtubules), neuron signaling, low concentration biomolecule detection, biochemical reactions within the neuron, and the formation of lamellipodia and filopodia [1,2,7,21]. The resultant growth cannot be predicted for individual neurons due to this stochastic-deterministic interplay, however, the growth dynamics for populations of neurons can be modeled by probability functions that satisfy a set of well-defined stochastic differential equations, such as Langevin and Fokker–Planck equations [10,15,19,20,22,23,24,25,26,27,28,29]. In previous work, we have shown that axonal dynamics on uniform glass surfaces is described by an Ornstein-Uhlenbeck (OU) process, defined by a linear Langevin equation and stochastic white noise [19,20]. We have also reported that neurons cultured on poly-D-lysine coated polydimethylsiloxane (PDMS) substrates with periodic parallel ridge micropatterns of spatial periodicity *d* (henceforth referred to as the pattern spatial period) grow axons parallel to the surface patterns. We have studied axonal growth as a function of time on these micropatterned surfaces and found that axonal alignment increases as a function of time [16]. The axonal dynamics are described by non-linear Langevin equations, involving quadratic velocity terms and non-zero coefficients for the angular orientation of the growing axon [20]. In another paper, we have used the Langevin and Fokker–Planck equations to quantify axonal growth on surfaces with ratchet-like topography (asymmetric tilted nanorod: nano-ppx surfaces) [10]. We have shown that the axonal growth is aligned with a preferred spatial direction as a result of a “deterministic torque” that drives the axons to directions determined by the substrate geometry. We have also measured the angular distributions and the coefficients of diffusion and angular drift on these substrates [10]. Our results provide a detailed analysis of axonal growth on substrates with different geometrical patterns by measuring speed and acceleration distributions as a function of substrate geometry [20], axonal alignment as a function of time [16], as well as axonal angular distributions, angular drift, and diffusion coefficients [10,16,20]. However, a comprehensive model of the growth dynamics on these substrates, which takes into account the mechanisms of axonal alignment and cell-surface interactions, is still missing.

In this paper, we develop a discrete quantitative model of growth cone motility that incorporates the neuron’s ability to sense the substrate geometry. We demonstrate that the motion of axons on surfaces with micropatterned periodic geometrical patterns is governed by a feedback control mechanism that leads to axonal alignment on these surfaces. This theoretical model fully accounts for the experimental data measured on ensembles of axons, including speed distributions and angular alignment. Furthermore, our experiments show that the inhibition of cytoskeletal dynamics by treatment of neurons with Taxol (inhibitor of microtubules) and Y-27632 (inhibitor of myosin II and actin dynamics) results in a significant decrease in the axonal alignment by altering the feedback loop mechanism of the neuron. Our results demonstrate that axonal dynamics are controlled by a contact–guidance mechanism, which stems from cellular feedback imparted by the high-curvature geometrical features of the growth substrate. This work provides new insights for creating biomimetic systems that emulate neuronal growth in vivo, and it has a significant impact on designing new platforms for guiding the growth and regeneration of neurons.

## 2. Materials and Methods

The cells used in this work are cortical neurons obtained from embryonic day 18 rats. For cell dissociation and culture, we have used established protocols presented in our previous work [10,15,16,19,20,30,31,32]. In our previous work, we have performed immunostaining experiments that show high neuron cell purity in these cultures [30]. Cortical neurons were cultured on micropatterned polydimethylsiloxane (PDMS) substrates coated with poly-d-lysine (PDL). The cells were cultured at a surface density of 4000 cells/cm^2^. We have previously reported that neurons cultured at relatively low densities (in the range 3000–7000 cells/cm^2^) are optimal for studying axonal growth on surfaces with different mechanical, geometrical, and biochemical properties [10,15,16,19,20,30,31,32].

The periodic micropatterns on PDMS surfaces are made of parallel ridges separated by troughs. Each surface is characterized by a different value of the pattern spatial period *d*, defined as the distance between two neighboring ridges (Figure 1a). To make these periodic patterns, we used a simple fabrication method based on imprinting diffraction grids with different grating constants onto PDMS substrates (Appendix A and [33]). The direction of the pattern is shown in Figure 1 by the parallel bright stripes (ridges) and by the parallel dark stripes (troughs).

The micropatterned surfaces were spin-coated with a PDL (Sigma-Aldrich, St. Louis, MO, USA) solution of concentration 0.1 mg/mL. Both growth surfaces and the neuronal cells were imaged using an MFP3D atomic force microscope (AFM) equipped with a BioHeater closed fluid cell and an inverted Nikon Eclipse Ti optical microscope (Micro Video Instruments, Avon, MA, USA). All surfaces were imaged with the AFM (a total of 12 different images). All neuronal cells have been imaged using fluorescence microscopy (a total of 18 images). Fluorescence images were acquired using a standard Fluorescein isothiocyanate -FITC filter: excitation: 495 nm and emission: 521 nm (details on acquiring the fluorescence images are provided in the Appendix A and [33]). For the experiments on chemically modified cells, we have treated the neurons with either: (1) Taxol (10 μM concentration) or (2) the chemical compound Y-27632 (10 μM concentration), which have been added to the neuron growth medium at the time of plating (Appendix A). Previous work has shown that a concentration of 10 μM of Taxol is very effective in suppressing the microtubule dynamics [10,21,30] and that 10 μM of Y-27632 is very efficient in disrupting actin polymerization and the formation of actin bundles, thus reducing traction forces between the neurons and the growth substrates [31].

### Data Analysis

Growth cone position, axonal length, and angular distributions have been measured and quantified using ImageJ (National Institute of Health). The displacement of the growth cone was obtained by measuring the change in the center of the growth cone position. To measure the growth cone velocities, the samples were imaged using fluorescence microscopy every Δ*t* = 5 min for a total period of 1 h per sample. The 5 min time interval between measurements was chosen such that the typical displacement $\Delta \vec{L}$ of the growth cone in this interval satisfies two requirements: (a) is larger than the experimental precision of our measurement (~0.1 μm) [19,20]; (b) the ratio $\Delta \vec{L}/\Delta t$ accurately approximates the instantaneous velocity $\vec{V}$ of the growth cone. The speed of the growth cone is defined as the magnitude of the velocity vector: $V\left(t\right)=\left|\vec{V}\left(t\right)\right|$, and the growth angle *θ*(*t*) is measured with respect to the *x*-axis (growth angle and the *x*-axis are defined in Figure 1b).

Experimental data (Figure 2 and Appendix A) shows that over a distance of ~20 μm, the axons can be approximated by straight line segments, with a high degree of accuracy. Therefore, to obtain the angular distributions for the growth angle *θ* (Figure 3 and Appendix A), we have tracked all axons using ImageJ and then partitioned them into segments of 20 μm in length, following the same procedure outlined in our previous work [16,20]. Next, we have recorded the angle that each segment makes with the *x*-axis (the schematic is shown in Figure 1b). The total range [0, 2π] of growth angles was divided into 18 intervals of equal size $\Delta \theta_{0}=\pi /9$ (Figure 3). To obtain the speed distributions (Figure 4 and Appendix A), the range of growth cone speeds at each time point was divided into 15 intervals of equal size $\left|\Delta {\vec{V}}_{0}\right|.$

## 3. Experimental Results

Cortical neurons are cultured on PDL-coated PDMS surfaces with parallel micropatterns (periodic parallel ridges separated by troughs). The surfaces differ by the value of the pattern spatial period *d*, defined as the distance between two neighboring ridges (Figure 1a). We analyze the growth of both untreated and chemically modified neuronal cells on surfaces with spatial periods in the range *d =* 1 to 6 μm (in increments of 1 μm).

Figure 2a,b show examples of images of axonal growth for untreated neurons cultured on PDL-coated PDMS micropatterned surfaces with pattern spatial period: *d =* 3 (Figure 2a), and *d =* 5 μm (Figure 2b). We have previously demonstrated that axons of untreated neurons display maximum alignment along PDMS patterns for surfaces where the pattern spatial period *d* matches the linear dimension of the growth cone *l*, where *l* is in the range 2 to 6 μm [20]. The experimental data shown in Figure 2a,b is in agreement with our previous findings. Examples of the corresponding axonal normalized angular distributions are shown, respectively, in Figure 3a,b.

To further investigate the axonal dynamics on PDMS surfaces with periodic micropatterns, we measure the angular and speed distributions for neurons treated with chemical compounds known to inhibit the dynamics of the cell cytoskeleton. Figure 2c,d shows examples of axonal growth for neurons treated with 10 μM of Taxol and cultured on surfaces with pattern spatial period *d =* 3 (Figure 2c), and *d =* 5 μm (Figure 2d). Appendix A shows similar images obtained for neurons treated with 10 μM of Y-27632. All images for untreated, as well as chemically modified neurons, are captured at *t =* 36 h after cell plating.

Taxol is a chemical compound that is commonly used to inhibit the normal functioning of the cytoskeleton due to the disruption of microtubule dynamics [10,21,30]. Y-27632 is a chemical compound known to inhibit the formation of actin bundles and the reorganization of actin-based structures during neuronal growth [31,34]. Both of these compounds have been shown to be effective at the concentration of 10 μM used in our experiments [10,21,30,31,34]. The corresponding normalized angular distributions for axonal growth of Taxol modified neurons are shown in Figure 3c,d. Examples of growth images, as well as axonal angular and speed distributions for neurons treated with Y-27632, are shown in Appendix A. The neurons treated with either Taxol or Y-27632 show a dramatic decrease in the degree of alignment with the surface patterns compared to the unmodified cells (Figure 2 and Figure 3, Appendix A). The data show that while the axonal directionality is greatly reduced by the chemical treatment, the treated neurons still grow long axons and form cell–cell connections (Figure 2c,d and Appendix A). These results demonstrate that the disruption of the cytoskeletal dynamics for chemically treated neurons affects only the degree of alignment with the surface pattern, leaving the navigation of the growth cone and axonal outgrowth uninhibited.

Figure 4 (as well as Appendix A) shows that the speed distributions for both untreated and chemically modified growth cones are close to Gaussian distributions. This is indeed expected for culture times *t =* 36 h after cell plating, as shown in our previous work [16].

## 4. Theoretical Model for Axonal Dynamics

Axonal dynamics on the PDMS substrates is characterized by both deterministic and stochastic components [10,16,19,20,23]. The angular motion of axons on patterned PDMS surfaces is described by the growth angle *θ* defined in Figure 1 and Figure 2. In our previous work, we have shown that the probability distribution p(θ,t) for the growth angle satisfies the following Fokker–Planck equation [16]:(1)$$\frac{\partial }{\partial t}p\left(\theta ,t\right)=\frac{\partial }{\partial \theta }\left[-\gamma_{\theta }\cdot \cos\;\theta \left(t\right)\cdot p\left(\theta ,t\right)\right]+D_{\theta }\cdot \frac{\partial^{2}}{\partial \theta^{2}}p\left(\theta ,t\right)$$
where *D**_θ_* represents the effective angular diffusion (cell motility) coefficient, and $\gamma_{\theta \;}\cdot \cos\;\theta \left(t\right)$ corresponds to a “deterministic torque” representing the tendency of the growth cone to align with the preferred growth direction imposed by the surface geometry [16]. The stationary solution of Equation (1) is given by [16]:(2)$$p\left(\theta \right)=A\cdot \exp\left(\frac{\gamma_{\theta }}{D_{\theta }}\cdot \left|sin\;\left(\theta \right)\right|\right)$$
where *A* is a normalization constant obtained from the normalization condition:$$\int_{0}^{2\pi }p\left(\theta \right)\cdot d\theta =1$$

The absolute value |sin *θ*| in Equation (2) reflects the symmetry of the growth around the *x*-axis: the angular distributions centered at θ=π/2 and θ= 3π/2 are symmetric with respect to the directions θ=π and θ=0 (Figure 3). This is a consequence of the fact that there is no preferred direction along the PDMS pattern (Figure 1 and Figure 2), and it applies to all types of micropatterned PDMS surfaces and for both untreated and chemically modified neurons. We also note that the deterministic torque has a maximum value if the growth cone moves perpendicular to the surface patterns (*θ =* 0 or *θ = π*), in which case the cell-surface interaction tends to align the axon with the surface pattern. The torque is zero for an axon moving along the micropattern.

The speed distribution p(V,t) of axonal growth is given by [16]:(3)$$\frac{\partial }{\partial V}p\left(V,t\right)=\frac{\partial }{\partial V}\left[\gamma_{s}\cdot \left(V-V_{s}\right)\cdot p\left(V,t\right)\right]+\frac{\sigma^{2}}{2}\cdot \frac{\partial^{2}}{\partial \theta^{2}}p\left(V,t\right)$$
where $\gamma_{s\;}$ is the constant damping coefficient of the corresponding Langevin equation ($\gamma_{s\;}=1/\tau$ where τ is the characteristic decay time), $V_{s}$ is the average stationary speed of the neuron population, and σ is noise strength for an uncorrelated Wiener process with Gaussian white noise [16,28,29]. Equation (3) has the following stationary solution [16]:(4)$$p\left(V\right)=B\cdot \exp\left(-\frac{\gamma_{s}}{\sigma^{2}}\cdot {\left(V-V_{s}\right)}^{2}\right)$$
where *B* is a normalization constant obtained from the normalization condition:$$\int_{0}^{\infty }p\left(V\right)\cdot dV=1$$

The model described by the Equations (1)–(4) predicts that the overall motion for the axons has two components: (a) a uniform drift along the directions of the PDMS micropattern (i.e., along the *y*-axis in Figure 1b), and a random walk around these equilibrium positions. This is indeed what is observed experimentally. At the beginning, the growth cone dynamics resemble a Brownian motion, resulting in a slow increase in the mean growth cone position along the *x*-axis. As time progresses, the axon exhibits a feedback control, which steers the axonal motion along the micropatterned parallel PDMS lines (Figure 1 and Figure 2). Furthermore, in the absence of the micropatterns, the motion for the growth cones reduces to a regular diffusion (Ornstein–Uhlenbeck) process characterized by an exponential decay of the autocorrelation functions with a characteristic time $\tau =\frac{1}{\gamma_{\theta }}$, axonal mean square length that increases linearly with time, and velocity distributions that approach Gaussian functions [28,29]. In our previous work, we have shown that this is indeed the case for axonal growth on PDL coated glass and PDMS surfaces characterized by large pattern spatial periods: *d* > 9 μm [16,20].

We use Equations (1)–(4) to fit the normalized experimental angular and speed distributions for each type of surface and cell (untreated or chemically modified) considered in our experiments (fits to the data are represented by the continuous red curves in Figure 3 and Figure 4, Appendix A). For the case of untreated neurons, the theoretical model fits the experimental data for the angular probability distributions with the following values of the deterministic torque: $\gamma_{\theta }=\left(0.16\pm 0.02\right){\;h}^{-1}$ (for growth on surfaces with *d =* 3 μm, Figure 3a), and: $\gamma_{\theta }=\left(0.18\pm 0.03\right)\;h^{-1}$ (for growth on surfaces with *d =* 5 μm, Figure 3b). Similarly, for neurons treated with Taxol, we obtain from the data fit: $\gamma_{\theta }=\left(0.11\pm 0.02\right){\;h}^{-1}$ (for growth on surfaces with *d =* 3 μm, Figure 3c), and: $\gamma_{\theta }=\left(0.13\pm 0.03\right){\;h}^{-1}$ (for growth on surfaces with *d =* 5 μm, Figure 3d). Values obtained for neurons treated with Y-27632 are shown in Appendix A. Appendix A also presents a summary of the values for the growth parameters obtained from the comparison between the theoretical model and the experimental data for different cells and substrates. We note that the values for the growth parameters decrease upon the chemical treatment of the neuron.

We use the solutions for the probability distributions given by the theoretical model presented above to simulate axonal growth trajectories, as well as axonal speed and angular distributions. The simulations are performed using the values for the angular diffusion coefficient and deterministic torque obtained from the fit to the experimental data (Figure 3 and Figure 4, Appendix A) with no additional adjustable parameters (see Appendix A for simulation details). Figure 5 shows examples of simulation results for untreated (Figure 5a,b) and Taxol-treated neurons (Figure 5c,d). Similar simulations obtained for neurons treated with Y-27632 are shown in Appendix A.

We emphasize that the angular distributions and speed distributions obtained from these simulations match the experimental data for untreated, Taxol treated, and Y-27632 treated neurons without the introduction of any additional parameters. For example, the simulated axon trajectories in Figure 5a,b reproduce the high degree of alignment observed experimentally for untreated neurons grown on surfaces with *d =* 3 or *d =* 5 μm (Figure 2a,b and Figure 3a,b). Figure 5c,d and Appendix A show simulated growth trajectories with an intermediate degree of alignment (similar to the data measured on Taxol and Y-27632 treated neurons in Figure 2c,d and Figure 3c,d, Appendix A).

## 5. Feedback Mechanism for Axonal Growth

The theoretical model and the simulations presented in the previous section imply that the axonal motion on surfaces with periodic geometries exhibits a simple closed-loop “automatic controller” behavior: the growth cone detects the geometrical cues on the surface and tends to align its motion along certain preferred directions that maximize the cell-surface interactions. In general, feedback control means that the system is steered towards a target behavior using information that is retrieved from the environment through continuous measurements. This is a powerful technique for describing the dynamical properties of many types of physical and biological systems, including particle trapping [35,36,37], optical tweezers [38,39,40], neuron firing [41,42], and cellular dynamics [43,44,45].

To further investigate the automatic controller model, we measure the variation of the deterministic torque ($\gamma_{\theta }$) (control parameter) with the pattern spatial period (*d*) (external stimulus). Figure 6 shows the variation of the experimentally measured values for $\gamma_{\theta }$ with *d*, for untreated neurons (red squares), as well as for neurons treated with Taxol (black squares) and Y-27632 (green squares). As shown in references [43,44,45] on work performed for galvanotaxis and chemotaxis dose-response curves for the motion of human granulocytes and keratinocytes, the automatic controller model leads to the following dependence for the variation of the control parameter with the external stimulus:(5)$$\gamma_{\theta }\approx \frac{I_{1}\left(\beta \cdot d\right)}{I_{2}\left(\beta \cdot d\right)}$$
where *I*_1_ and *I*_0_ are the modified Bessel functions of the first kind, and β is a parameter with dimensions of inverse length. The dotted curves in Figure 6 represent fits to the data with the predictions of the closed-loop feedback model given by Equation (5).

The data in Figure 6 demonstrate that axonal dynamics on micropatterned PDMS surfaces is described by a simple automatic controller model with a linear response, when the pattern spatial period is in the range *d =* 1–6 μm, which is when *d* matches the linear dimension of the growth cone: d≈l. This conclusion applies to both untreated cells and cells treated with Taxol and Y-27632. In all these cases, the pattern spatial period (*d*) plays the role of an effective external (geometric) stimulus that determines the axonal alignment, similar to the electric field in the case of galvanotaxis of human granulocytes and keratinocytes [22,43], or the concentration gradient in the case of cellular chemotaxis [44]. Furthermore, Figure 6 demonstrates that the response of the automatic controller is affected by the inhibition of cytoskeletal dynamics: the actual response (measured by the coefficient β) is different for the untreated and chemically treated cells (see the caption in Figure 6, and Appendix A).

## 6. Discussion

It is well-known that neurons respond to a variety of external cues (biochemical, mechanical, geometrical) while wiring up the nervous system in vivo [1,2,4,5,6,7]. In many cases, these cues consist of periodic geometrical patterns with dimensions in the order of a few microns [2,4,5]. Our studies show that growth substrates containing micropatterned periodic features promote axonal growth along the direction of the pattern. The range for the micropattern spatial periods in our experiments (*d =* 1–6 μm) is relevant both for neuronal growth in vivo, as well as, for many proposed biomimetic implants for nerve regeneration [9,14]. Our experiments show that neurons grown on PDMS substrates display a significant increase in the overall axonal length and a high degree of alignment when the pattern spatial period (*d*) matches the linear dimension of the growth cone: d ≈ l.

In this paper we demonstrate that the dynamics of the growth cones on surfaces with micropatterned periodic features are described by Fokker–Planck equations that capture all the characteristics of axonal growth for untreated and chemically modified neurons, including diffusion (cell motility) coefficients, angular, and speed distributions (Figure 1, Figure 2, Figure 3, Figure 4 and Figure 5 and Appendix A). Furthermore, this model implies a simple closed-loop “automatic controller” model for axonal motion: the growth cone detects geometrical features on the substrate and orients its motion in the directions that maximize the interaction between the axon and the substrate. Models based on the theory of automatic controllers have been successfully used by other groups to characterize the galvanotaxis (motion in external electric fields) of human granulocytes and keratinocytes [22,43], as well as the chemotactic response of bacteria and of various types of virus modified cells [44,45]. Our results show that the closed-loop feedback control underlies the mechanism of axonal alignment on micropatterned PDMS substrates. This behavior is displayed by both untreated and chemically modified neurons, as shown in Figure 6. In this figure, each data set (for untreated, Taxol, and Y-27632 treated cells) is fitted with a unique parameter β, which demonstrates a linear response characteristic to a proportional controller: the response is proportional to the signal received from the guidance cue [43,44]. The coefficients β obtained from the data fit (Figure 6 and Appendix A) measure the neuronal responses to periodic geometrical cues and play a similar role to the galvanotaxis and chemotaxis coefficients used to describe the cellular motion in external electric fields or chemical gradients [22,43,44,45]. Within this model, the growth cone behaves similarly to a “device” that senses geometrical cues, and as a result, generates traction forces that align the axon with the surface pattern.

These results support our previous findings that neurons follow geometrical patterns through a contact–guidance mechanism [10,20]. Contact guidance is the ability of cells to change their motion in response to geometrical cues present in the surrounding environment. This behavior has been observed for several types of cells, including neurons, fibroblasts, and tumor cells [10,14,20,46]. Previous work [14,46,47,48] has shown that growth cones develop several different types of curvature sensing proteins (such as amphipathic helices and bin-amphiphysin-rvs (BAR)-domains) that act as sensors of geometrical cues and are involved in the generation of traction forces. Moreover, the degree of directional alignment of cellular motion is increasing with the increase in the density of curvature sensing proteins [47,48]. In our experiments, the growth cone filopodia and lamellipodia wrap around the ridges of the PDMS micropatterns [16], which results in a minimal contact area with the surface, and thus a maximum density of curvature sensing proteins. Consequently, high-curvature geometrical features such as ridges on PDMS substrates will impart higher forces to the focal contacts of filopodia wrapped over these features, compared to the low-curvature patterns. This means that the contact guidance mechanism leads to an increase in the traction force along the direction of the surface pattern (defined as the *y*-direction in Figure 1b), which ultimately results in the observed directional alignment of axons on these surfaces.

Both microtubules and actin filaments inside the growth cone act as stiff load-bearing structures that generate traction forces [1,2,4]. Inhibition of microtubule or actin dynamics will therefore result in a decrease in cell-substrate interactions. Our experiments demonstrate that disruption of the cytoskeletal dynamics for cells treated with Taxol (inhibitor of microtubule dynamics) and Y-27632 (disruption of actin filaments) results in a decrease in the degree of alignment and a reduction in cell-substrate interactions (Figure 2c,d, Appendix A). Furthermore, the smaller values of the parameter β for the chemically treated neurons imply a less effective guidance mechanism for these cells compared to the untreated ones. Thus, the results obtained in the case of chemically treated neurons show an alteration of the automatic controller responsible for the directional motion of axons. These experimental results are in agreement with the predictions of the contact guidance mechanisms discussed above.

The automatic controller model presented in this paper could be further extended to account for the explicit dependence of the growth parameters on the mechanical and biochemical guidance cues, such as changes in the stiffness of the growth substrate or external chemical gradients. This approach could provide significant insight into the neuronal response to external stimuli without the need to incorporate all the molecular steps involved. The model discussed here could also be applied to other types of cells to give new insight into the nature of cellular motility. In future experiments, the staining neurons with specific markers will allow us to identify the morphological components of the growth cone (lamellipodia, filopodia) and the distribution of cytoskeletal components (actin filaments, microtubules) inside the cell. These experiments will also involve the measurement of both cell-surface coupling forces using traction force microscopy and the density of cell surface receptors and curvature sensing proteins using high-resolution fluorescence techniques. In principle, these future investigations will enable researchers to quantify the influence of environmental cues (geometrical, mechanical, biochemical) on cellular dynamics and to relate the observed cell motility behavior to cellular processes, such as cytoskeletal dynamics, cell-surface interactions, and signal transduction.

## 7. Conclusions

In this paper, we have performed a detailed experimental and theoretical analysis of axonal growth on micropatterned PDMS surfaces. We have demonstrated that the axonal dynamics on these surfaces are described by a theoretical model based on the motion of a closed-loop automatic controller in a substrate with periodic geometrical features. We have used this model to measure the growth parameters that characterize the axonal motion. Our results show that the dynamics of the growth cone are regulated by a contact–guidance mechanism, which stems from cellular feedback in an external periodic geometry: the growth cone responds to geometrical cues by rotating and aligning its motion along the surface micropatterns. The general model presented here could be applied to describe the dynamics of other types of cells in different environments, including external electric fields, substrates with various mechanical properties, and biomolecular cues with different concentration gradients.

## Figures and Tables

**Figure 1 biomimetics-06-00041-f001:**
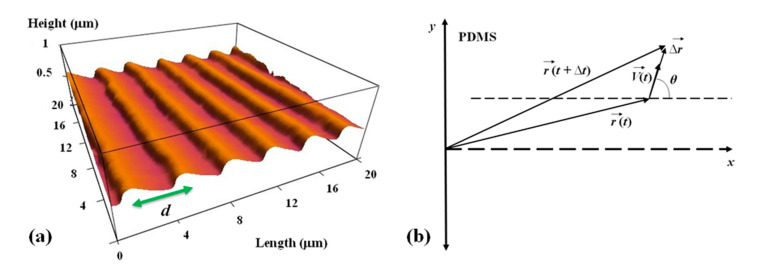
(**a**) Atomic Force Microscope (AFM) topographic image of a PDL coated PDMS patterned surface. The image shows that the micropatterns are periodic in the *x*-direction with the spatial period *d =* 4 μm and have a constant depth of approximately 0.5 μm. (**b**) Coordinate system and the definition of the angular coordinate *θ*. The *x*-axis is defined as the axis perpendicular to the direction of the PDMS patterns.

**Figure 2 biomimetics-06-00041-f002:**
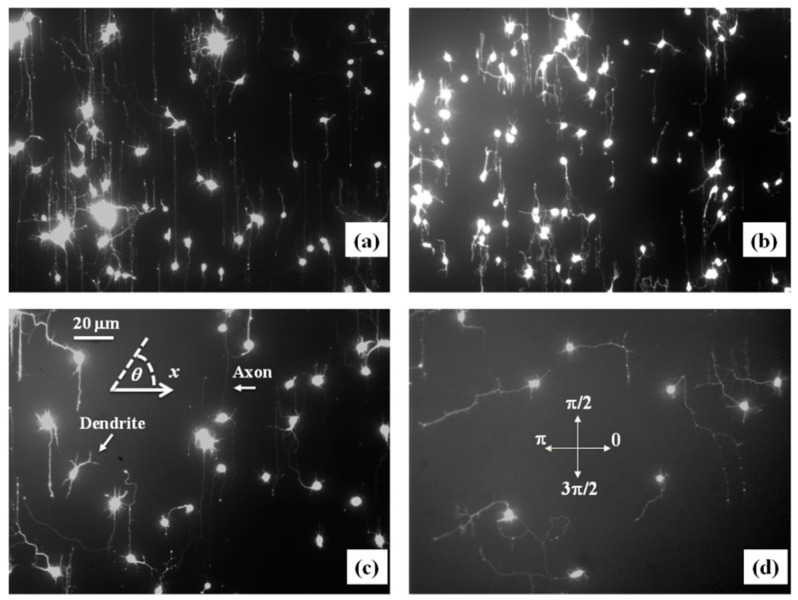
Fluorescence (Tubulin Tracker Green) images showing examples of axonal growth for cortical neurons cultured on PDL-coated PDMS surfaces with periodic micropatterns. (**a**,**b**) Examples of growth for untreated cortical neurons grown on PDMS substrates with pattern spatial period: *d =* 3 μm in (**a**), and *d =* 5 μm in (**b**). (**c**,**d**) Examples of axonal growth for cortical neurons treated with Taxol, a chemical compound that inhibits the microtubule’s dynamics. The pattern spatial period is *d =* 3 μm in (**c**), and *d =* 5 μm in (**d**). The main structural components of a neuronal cell are labeled in (**c**). Cortical neurons typically grow in a long process (the axon) and several minor processes (dendrites). The axon is identified by its morphology, and the growth cone is identified as the tip of the axon. The angular coordinate *θ* used in this paper is defined in (**c**). The directions corresponding to θ=0, π/2, π, and 3π/2 are shown in (**d**). All angles are measured with respect to the *x*-axis, defined as the axis perpendicular to the direction of the PDMS patterns (see Figure 1b). All images are captured 36 h after neuron plating. The scale bar shown in (**c**) is the same for all images.

**Figure 3 biomimetics-06-00041-f003:**
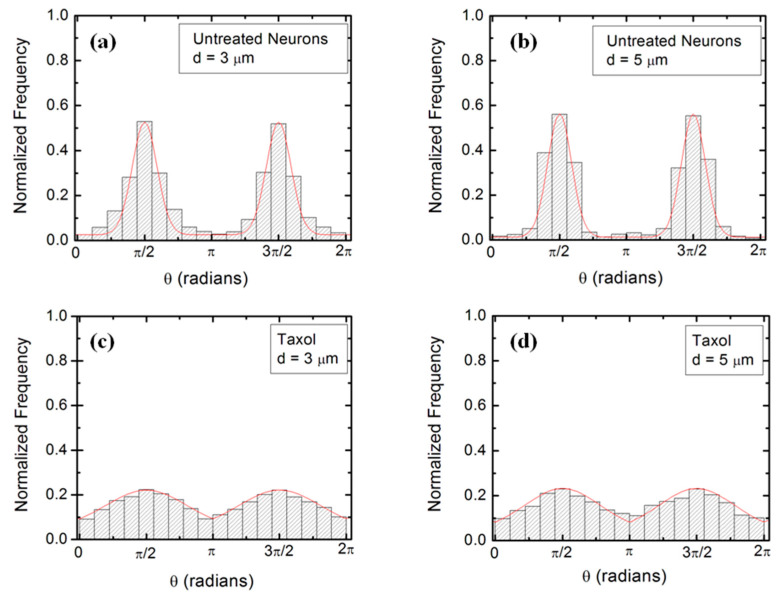
Examples of normalized experimental angular distributions for axonal growth for neurons cultured on micropatterned PDMS surfaces with different pattern spatial periods *d*. The vertical axis (labeled normalized frequency) represents the ratio between the number of axonal segments growing in a given direction and the total number, N, of axon segments. Each axonal segment is 20 μm in length (see Section 2 on Data Analysis). All distributions show data collected at *t* = 36 h after neuron plating. The continuous red curves in each figure are the predictions of the theoretical model discussed in Section 4. (**a**) Angular distribution obtained for *n* = 1240 different axon segments (350 axons) for untreated neurons cultured on surfaces with *d =* 3 μm (corresponding to Figure 2a). (**b**) Angular distribution obtained for *n* = 1158 (324 axons) different axon segments for untreated neurons cultured on surfaces with *d =* 5 μm (corresponding to Figure 2b). The data shows that the axons display strong directional alignment along the surface patterns (peaks at θ=π/2 and θ=3π/2), with a high degree of alignment given by the sharpness of the distributions. (**c**) Angular distribution obtained for *n* = 1093 different axon segments for neurons treated with Taxol and cultured on surfaces with *d =* 3 μm (corresponding to Figure 2c). (**d**) Angular distribution obtained for *n* = 845 different axon segments for neurons treated with Taxol and cultured on surfaces with *d =* 5 μm (corresponding to Figure 2d). The neurons treated with Taxol show a significant decrease in the degree of alignment with the surface patterns compared to the untreated cells.

**Figure 4 biomimetics-06-00041-f004:**
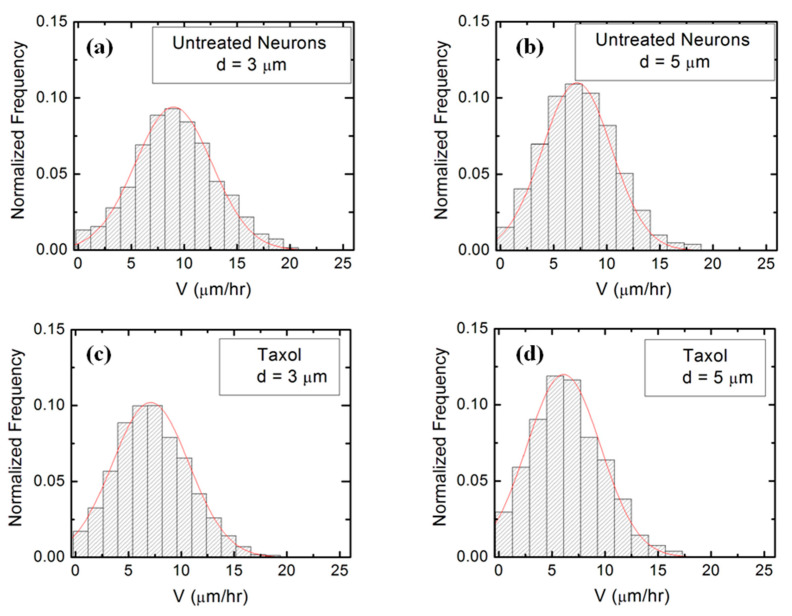
Examples of normalized speed distributions for growth cones measured on micropatterned PDMS surfaces with different pattern spatial period *d*. All distributions show data collected at *t* = 36 h after neuron plating. The continuous red curves in each figure are data fits with the theoretical model discussed in Section 4. (**a**) Speed distribution for *n* = 452 different growth cones measured for untreated neurons cultured on surfaces with *d =* 3 μm. (**b**) Speed distribution for *n* = 488 different growth cones measured for untreated neurons cultured on surfaces with *d =* 5 μm. (**c**) Speed distribution for *n* = 416 different growth cones measured for Taxol treated neurons cultured on surfaces with *d =* 3 μm. (**d**) Speed distribution for *n* = 395 different growth cones measured for Taxol treated neurons cultured on surfaces with *d =* 5 μm.

**Figure 5 biomimetics-06-00041-f005:**
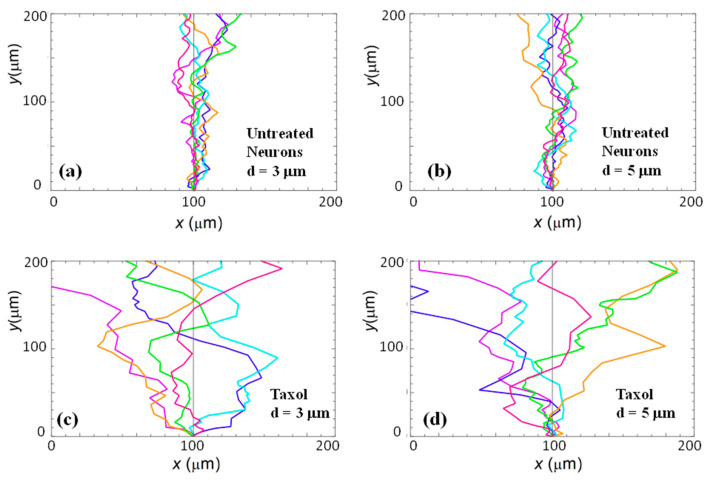
Examples of simulated neuronal growth for: untreated (**a**,**b**); and Taxol-treated (**c**,**d**) neurons. The simulations are performed using the values of the growth parameters obtained from the fit of the experimental data with Equations (2) and (4) (see main text). The pattern spatial periods correspond to the data shown in Figure 3 and Figure 4: *d =* 3 μm for (**a**,**c**), and *d =* 5 μm for (**b**,**d**).

**Figure 6 biomimetics-06-00041-f006:**
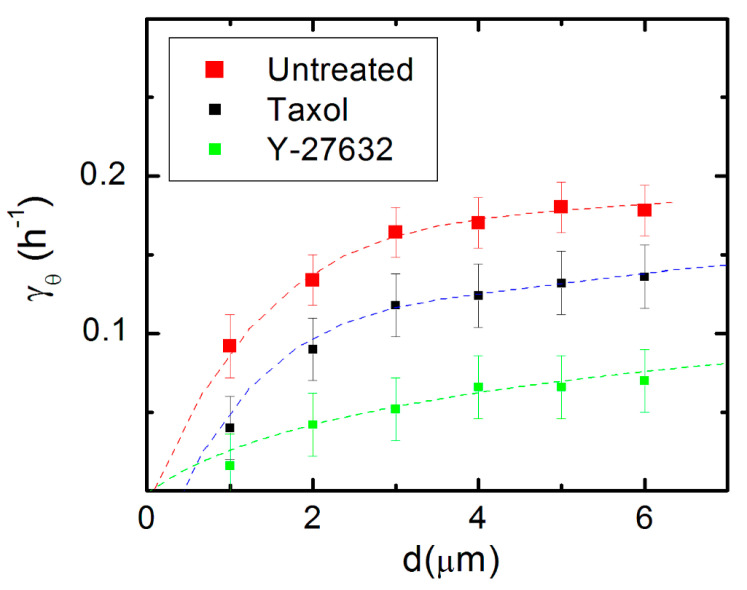
Variation of the deterministic torque ($\gamma_{\theta }$) (control parameter) with the pattern spatial period (*d*) (external stimulus) for axonal growth on micropatterned PDMS substrates. The red squares represent the values for $\gamma_{\theta }\;$ obtained from the fit to experimental data for untreated neurons. The black squares correspond to the experimental data obtained for neurons treated with Taxol, while the green squares correspond to the data measured for neurons treated with Y-27632. Error bars indicate the standard error of the mean for each data set. The dotted curves represent the fit of the data points with Equation (5). The graph shows that data points in this range are fitted by the feedback control model for the following values of the parameters: $\beta =\left(1.2\pm 0.4\right)\;{\mu m}^{-1}$ for untreated neurons, $\beta =\left(0.6\pm 0.3\right)\;{\mu m}^{-1}$ for neurons treated with Taxol, and $\beta =\left(0.4\pm 0.3\right)\;{\mu m}^{-1}$ for neurons treated with Y-27632.

## Data Availability

The data presented in this study are available within the manuscript and its Appendix A.

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
