# Peer review of "Neuronal Growth and Formation of Neuron Networks on Directional Surfaces"

_biomimetics, 2021, doi:10.3390/biomimetics6020041_

Round 1
Reviewer 1 Report
This is a fundamental study investigating the relationship between axonal dynamics and period geometric features, rationalizing the result with a biophysical model. The discussion is solid, and the cited literature is appropriate.
I have a few questions regarding the experimental part that need to be addressed before publication.
What is the rationale behind collecting data by means of both AFM and fluorescence imaging for collecting the data discussed in this study?
It is not clear how many AFM and fluorescence images have been respectively collected to derive the data.
On page 7 the Authors say that “neuronal cells were imaged using an MFP3D atomic force microscope (AFM) equipped with a BioHeater closed fluid cell, and an inverted Nikon Eclipse Ti optical microscope”, while later (page 8), they say “To measure the growth cone velocities the samples were imaged every Dt = 5 min for a total period of 1 hr per sample”. In the case of repeated AFM imaging being performed on the same neurons, it is not clear if the imaging process can damage the cells, and to what extent. Also, what controls have been implemented to clarify/quantify this aspect?
What are the images shown in Fig 2? Are they AFM images or fluorescence? Same problem with Figure S1 (in the SI).
In Fig 3 caption, the Author states that the data are obtained from “N = 1240 different axon segments for untreated neurons cultured on surfaces with d =3 μm”, and “N = 1158 different axon segments for untreated neurons cultured on surfaces with d =5 μm”, while it is not clear how many neurons have been imaged and how many images have been produced. Also, again it is not clear if they are AFM or fluorescence images or both.
It would be useful for the Authors to discuss the throughput of the imaging method, and perhaps describe its possible limitations and ways to improve it towards more general applicability of their technique.
Reviewer 2 Report
In the present manuscript Yurchenko et al. described a method to study and predict the neurite outgrowth of primary neurons. The study sounds solid and well contextualized in the field. However, different missing points need to be addressed in the biological point of view.
Major revisions:
- In the cell culture preparation protocol, it is not explained if these cultures are pure neuronal or mixed with astrocytes. If no astrocytes inhibitors (such as Ara-C) is added to the culture, than a mixed neurons/astrocytes population is obtained.
A nuclear staining should be added to the pictures to visualize the percentage of neurons compared to non-neuronal cultures.
- Cultures treated with both Taxol and Y-27632 look, not only morphologically, but also different in cell density. Is there any effect on cell viability? How the influence on cell viability impact on the analysis? For a morphological analysis a non-toxic dose should be used.
- When neurons are cultured, it is not possible to discriminate between axons and dendrites, especially with only a tubulin marker. In fact, in neuronal cultures, are usually indicated with the general term “neurites”. If the axon/dendrite discrimination is needed, then specific axonal/dendritic markers should be used.
- The same generalization issue emerges different times throughout the paper. It is not possible to discuss about, growth cone, filopodia or lamellipodia, without using specific markers for these structures. Moreover, a low magnification imaging, as showed in the paper, is not able to discriminate these structures by using only tubulin markers.
Minor revisions:
- Please indicate in the figure the staining used to visualize the cells.
- Since the two molecules, Taxol and Y-27632, are discussed with the same weight in all the sections throughout the manuscript, Y-27632 images and data should be added in the main text.
